# Biomechanics Perspective for the Design and Manufacture of Continuous Ambulatory Peritoneal Dialysis Connectors

Mario Alberto Grave-Capistrán [1], Luis Antonio Aguilar-Pérez [1], Juan Carlos Paredes-Rojas [2], Carlos De la Cruz-Alejo [3] and Christopher René Torres-SanMiguel [1,*]

1   Instituto Politécnico Nacional, Escuela Superior de Ingeniería Mecánica y Eléctrica, Sección de Estudios de Posgrado e Investigación Unidad Zacatenco, Ciudad de México 07738, Mexico; mgravec1200@alumno.ipn.mx (M.A.G.-C.); laguilarp@ipn.mx (L.A.A.-P.)
2   Centro Mexicano para la Producción más Limpia, Instituto Politécnico Nacional, Acueducto de Guadalupe S/N, La laguna Ticomán, CP. 07340, Ciudad de México 07738, Mexico; jparedes@ipn.mx
3   Instituto Politécnico Nacional, Escuela Superior de Ingeniería Mecánica y Eléctrica, Unidad Culhuacán, 04440, Ciudad de México 07738, Mexico; cdelacruza@ipn.mx
*   Correspondence: ctorress@ipn.mx; Tel.: +52-555-729-600 (ext. 54815)

**Featured Application: These connectors are used in continuous ambulatory peritoneal emergency dialysis for exchanging different dialysis bags and transfer line brands.**

**Abstract:** Chronic kidney disease (CKD) progressively and irreversibly affects the kidneys and is considered a catastrophic disease on a global scale. Continuous ambulatory peritoneal dialysis (CAPD) is one of the most used methods of treatment, involving infusion and draining bags and a transfer line. Patients receiving this treatment have a catheter that is placed during surgery; this depends on having the same catheter supplies, as they are not compatible with other market brands. This research shows the comparison between connector brands used at the outlet of the Tenckhoff® catheter. The methodology shows the design of two connectors using the 3D printing technique. Numerical simulations were carried out to establish the flow patterns through each of the designs; the maximum values of velocity reached 74 mm/s inside the PISA to Baxter (PB) connector, while the pressure and vorticity were controlled and did not represent failures inside the connectors and threads connections. An experimental testbed was designed to verify the connections between the manufactured devices and the market brand elements. The results show numerical and experimental comparisons of the developed titanium-ELI connectors with no leaks at the connection points due to the lack of commercial supplies. These connectors can be used in the treatment of CAPD.

**Keywords:** 3D printing; chronic kidney disease; connectors; numerical simulation; testbed

## 1. Introduction

Chronic kidney disease (CKD) is a degenerative and irreversible chronic disease. The last decade has seen an increased rate in different countries, establishing it as a catastrophic disease. Mexico already faces a severe crisis with kidney disease issues, particularly regarding the economic implications that face the national health industry. Additionally, the number of patients has exceeded the sector's capacity [1,2]. CKD is classified according to the guidelines, Kidney Disease: Improving Global Outcomes (KDIGO) in five phases, which are each characterized by their glomerular filtration rate (GFR) as well as their appropriate level of treatment, starting with diagnoses to delay the disease until the beginning reconstructive regressive therapy (RRT). Renal replacement therapies are used in phase V when the GFR is less than $15/mL/m^2$sc [3,4]. The acceptable range for GFR in men is around 130 mL/min per 1.73 $m^2$ of body surface area and 120 mL/min in women; nevertheless, there is no set value taking into account the age, gender, and condition of the patient, although this can be estimated by some mathematical expressions [5].

In Mexico, the most common renal replacement therapy in medical institutions is peritoneal dialysis (PD), which has different modalities that use different techniques to comply with cleansing treatment for the human body. Automated peritoneal dialysis (APD) is an automatic machine treatment. The replacements are performed in a programmed and personalized way, allowing the patient freedom during the day since the therapy is carried out at night in an automatic and scheduled manner. This modality allows the use of a cycler device so that the patient can perform exchanges at home, heat the dialysate solution, drain the intra-abdominal cavity, and infuse a certain volume of the liquid. Additionally, it allows the patient to drain the effluent at a pre-programmed time and to repeat the process for the number of cycles indicated by their medical instructions. These devices have specific mechanical and hydraulic systems that effectively aid PD therapy and tightly control variables such as solution pressure and flow rate. APD has numerous advantages; one of them is that since the treatment occurs during the patient's sleep, a greater flow volume can be perfused, allowing better ultrafiltration with the same residence time as in continuous ambulatory peritoneal dialysis (CAPD), thus optimizing the number of exchanges. The use of DPA instead of CAPD is usually indicated in specific situations where the number of exchanges needs to be increased to achieve a relevant dose of dialysate fluid [6,7].

Continuous ambulatory peritoneal dialysis (CAPD) is the most common renal replacement therapy in the world. It is a prescription in which the patient is manually administered dialysate solution three or four times throughout the day [8]. The treatment uses system bags for the infusion, permanence, and drainage of liquid, exchanging fluid through a transfer line between the peritoneal cavity and the human body's treatment elements. It is a simple technique with manual daytime replacements in six-hour periods. Intermittent peritoneal dialysis (IPD) is a technique that is generally performed at the hospital and requires replacement dialysis infusion bags and intermittent draining with short-stay dialysate inside the peritoneum. It has a high risk of infection due to the multiple connections that are used for the treatment [9–11]. In CAPD, the system is connected by a transfer line with a catheter connecting a bag set that has specific features, according to the manufacturer. The transfer line is a fundamental piece of equipment that allows a connection between the peritoneal cavity catheter and the bag set; it can be made of titanium or plastic and features a protective cap to sterilize the transfer line [12]. It is essential that the commercial system used to carry out the treatment be followed; regarding the handling of the components, extreme caution must be taken, and rigorous hygiene practices must be used to achieve a correct connection between the elements and carry out the procedure satisfactorily. Substitute therapies for patients suffering from chronic kidney disease are currently widely accepted, and comply with treatment. There have been multiple attempts made by different authors to improve these rudimentary peritoneal dialysis systems; numerous types of tubes have been used, such as Foley catheters, catheters with particular geometries in the tip mushroom and whistle, pipe connectors polyethene rubber, glass, stainless steel, a double-lumen tube of Rosenak. Most of the connection systems showed leakage of the liquid and, consequently, infections in the peritoneal cavity. The first peritoneal dialysis technique performed by nurses was one named "Y" with a tube-shaped bottle that was no connected to the connector but was connected to the rudimentary dialysis catheter [13,14]. In the 1970s, plastic bags in the form of a container began to be used to transport dialysis fluid. Peritoneal dialysis became popular thanks to continuous ambulatory peritoneal dialysis (CAPD) by Popovich and Moncrief. In these years, the machines' operation was rudimentary, although several items were used to provide infusion and drainage to patients. Meanwhile, CAPD was the preferred dialysis technique since it did not require prominent investment or lengthy training to understand the process; it is still used as renal replacement therapy [15].

Nowadays, in Mexico, the Baxter® and PISA® companies are two different brands dedicated to providing support to connection elements for peritoneal dialysis. Connection systems for CAPD should ensure the safety of the patient by decreasing the incidence of infections due unclean handling. The methodology used in this work is based on

nephrology, and we developed a device through preliminary sketches. Through numerical simulations, we achieved a detailed design for developing a prototype in order to verify its functionality. Finally, prototypes with specific treatment characteristics were developed under medical-grade alloy and subjected to experimental tests. The materials used to carry out the present work are classified into three phases. The first one involves using the 3D prototyping technique in acrylonitrile butadiene styrene (ABS) to obtain a visual studio to establish improvements in the connectors. The second stage consists of manufacturing the detailed design in titanium "Ti-6Al-4V ELI", which has been used as a material in prostheses and stents and has received acceptance from the health sector [16]. Finally, in the third stage, an experimental testbed is built to evaluate the connection thread areas in equipment from the different brands already mentioned. Figure 1 shows a flowchart considering the methodology and stages for the design of each connector.

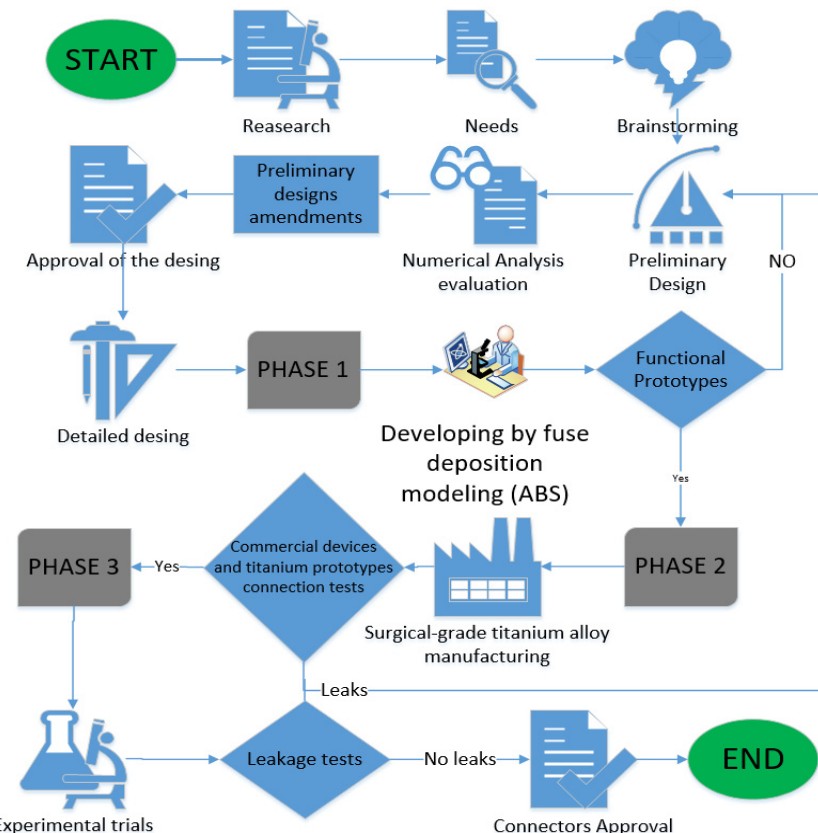

**Figure 1.** Luxury connector diagram for development.

Our aim is to establish a connection in the systems for CAPD in Mexico, mixing the resources to perform the treatment. In the materials and methods section, we show the design of the connectors for PD. Using the finite element method (FEM), we generated a virtual model simulation to analyze the fluid behavior through the designed connectors. Nowadays, the 3D printing technique allows one to obtain preliminary prototypes to improve the design in pieces in order to obtain detailed designs for production. In this work, 3D models using ABS material were used for preliminary connection tests with the commercial systems. Once the tests were correctly completed, two prototypes using titanium alloy were made to complement the research through an observational study. An experimental testbed was developed to verify leaks in each connection thread between the Baxter to PISA (BP) and PISA to Baxter (PB) connectors with the transfer lines and bag sets from commercial brands; the testbed worked as a support system, providing a means to accomplish the connection medical protocol. The results focus on evaluating the complex geometries and materials used in PD. The discussions and conclusions sections discuss an alternative to performing PD for emergency cases.

## 2. Connector Design

The connectors were designed in a Computer-Aided Design (CAD) program for modeling in 3D (SolidWorks® student version) and based on the applicable standards for threading. A technical sketch was obtained for both connectors, and we estimated the possibility of making a successful connection between the connectors and the commercial Baxter® and PISA® systems. The connectors are the main element that allows the dialysate solution from one brand's bag to the transfer line of another brand. Each connector has an inlet corresponding to the infusion and drainage bags and an outlet corresponding to the transfer line of each brand, allowing the combination of equipment from two different brands to carry out the exchange of the fluid for the CAPD treatment process. The connectors' development refers to industrial model designs that have their use in mechanics, specifically in the nephrology's area, as tool support for the treatment of CAPD. Connectors are characterized by their geometry and ornament, adapting a correct manipulation with the hands to make a successful connection attending the standards of Whitworth's thread giving a unique peculiarity that makes the difference and also provides a solution in emergency cases as a tool in patients which need the treatment avoiding medical surgeries changing the transfer line in use. Figure 2 represents the connection between the connectors developed and the commercial brands for the treatment of CAPD with the human body. It is noted that the connectors are the means to establish the use of a mixture of equipment from different brands (transfer line and infusion and drainage bags).

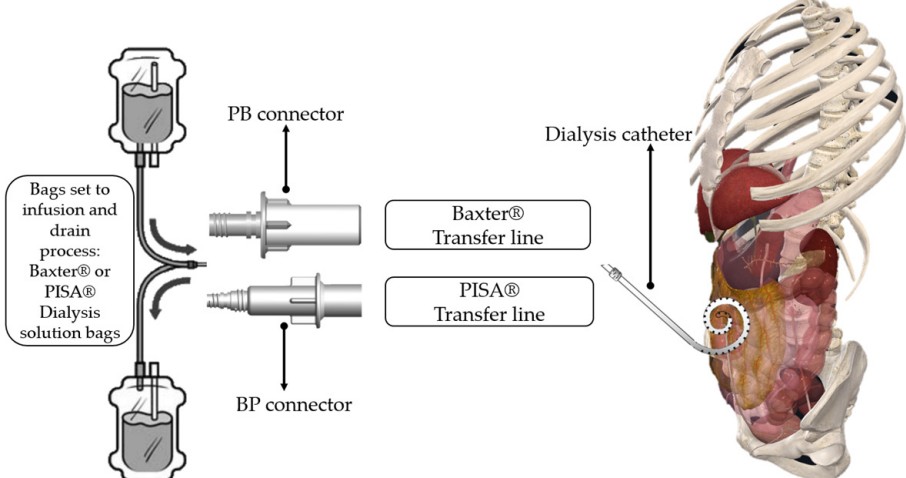

**Figure 2.** Comparison of the connectors for the different brands.

### 2.1. BP Connector

The BP connector allows a connection between the Baxter® infusion and drainage bags and the PISA® transfer line, allowing the dialysate fluid to be exchanged to carry out the treatment. The connector is divided into five main parts:

1. An inlet flow connection for Baxter® system bags;
2. A circumference limitation to avoid contamination of the nozzle connection between devices;
3. A BP connector malleability;
4. A cylindrical body of the BP connector;
5. A flow outlet connection for the PISA® transfer line.

In addition, the geometry and shape are followed by some essential characteristics:

- For handling: a special finger fastener geometry with a 1 mm round area for handling;
- For the fingers: a circular area with a diameter of 28 mm, the thickness of 2 mm, circular area at the top of 1 mm, and a circular area at the bottom of 0.50 mm;
- A mechanical seal to prevent fluid leakage;

- A suitable thread geometry connection for peritoneal Baxter® bags, towards the PISA® transfer line.

Figure 3 presents a diagram of the BP connector, showing a horizontal cross-sectional isometric view with a description of the main parts.

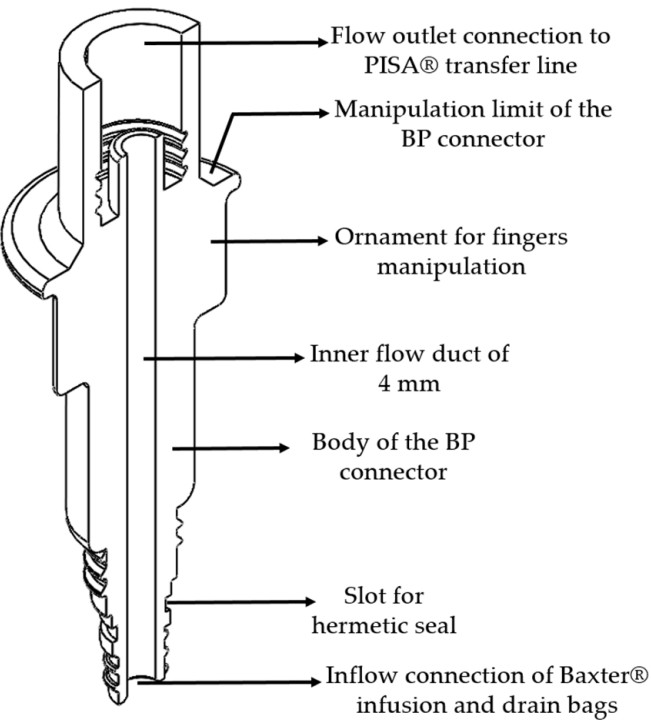

**Figure 3.** Isometric cross-section view of the BP connector.

### 2.2. PB Connector

The PB connector allows a connection from the infusion and drainage bags of the PISA® system to the transfer line of the Baxter® system, favoring the exchange of peritoneal fluid necessary to carry out peritoneal dialysis treatment. This connector has essential characteristics in terms of its internal conduit and input and output connection. Additionally, the connector characteristics are divided into five main parts:

1. An inlet flow connection for the PISA® bags system;
2. Circumference limitation to avoid contamination of the nozzle connection between devices;
3. A PB malleability connector;
4. A cylindrical body of the PB connector.
5. A flow outlet connection for the Baxter® transfer line.

Additionally, the geometry and ornament have specific characteristics, which are described below:

- For handling: special finger fasteners geometry with a chamfer and a 1 mm offset for handling with fingers;
- Internal duct reduction from 6.50 to 2.50 mm, through a chamfer of 2 mm with an inclination of 45°;
- For the fingers: circular area with a diameter of 28 mm, a thickness of 2 mm, circular area at the top of 1 mm, and circular area at the bottom of 0.50 mm;
- A mechanical seal to avoid fluid leakage;
- A suitable thread geometry connection of peritoneal PISA® bags to the Baxter®'s transfer line.

Figure 4 shows the technical drawing of the PB connector, shown in an isometric view and horizontal cross-section, as well as a description of its main parts.

The BP and PB connectors are designed to establish a successful connection between the commercial Baxter® and PISA® systems to favor emergency patient care. Both connectors have a slot in which to place a hermetic mechanical seal (o-ring) and prevent the leakage of the dialysis solution to ensure the connection between devices is airtight during the peritoneal dialysis treatment. The connector's shape geometry allows the correct manipulation of the device to avoid contamination that could cause peritonitis. The material used for manufacturing the connectors for the experimental tests is titanium "Ti-6Al-4VELI", which is used in medical devices and for the CAPD procedure; it can also be subjected to the cleaning process for reuse in the same treatment.

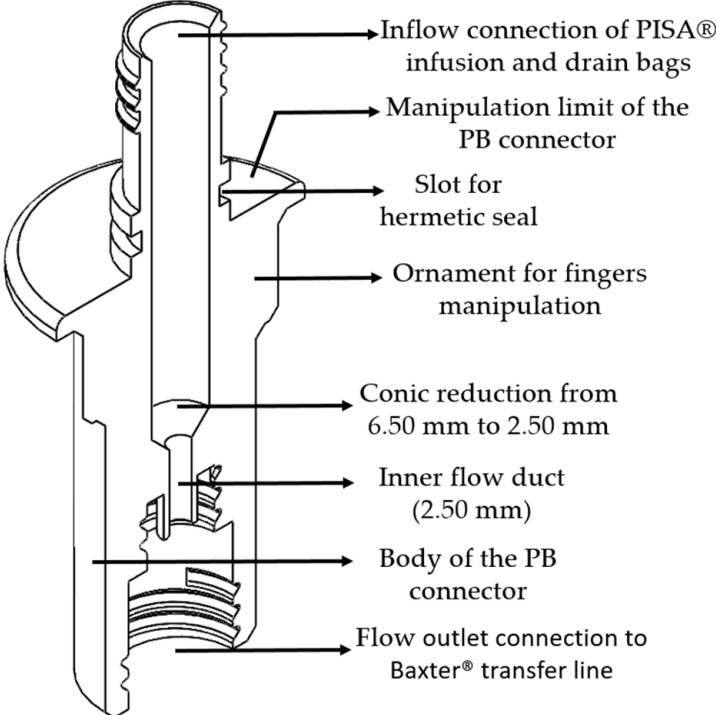

**Figure 4.** Isometric cross-section view of the PB connector.

### 3. Results

Three-dimensional prototypes were obtained using ABS in the Dimension SST 1200es printer. Table 1 shows the printer features and parameters, underlining the layer resolution used was 254 microns.

Three-dimensional models support the connection tests carried out under visual examination and allow us to establish each developed connector's correct functionality. The connection between these prototypes and the commercial devices used a detailed design to ensure that the thread connection and geometry were accurate for each connector. Additionally, 3D models in ABS were essential, as they allowed a visual analysis of the strategical manipulation in the medical sector, specifically in CAPD. Then, numerical simulations we performed considering the material properties. Figure 5 shows BP and PB connectors in ABS material made by a 3D printer.

**Table 1.** Dimensions of the SST 1200es printer.

| Brand | Stratasys |
|---|---|
| Machine type | 3D Printer |
| Technology | FDM (fused-deposition modeling) |
| Material modeling | ABSplus in ivory, white, black, red, green, olive, nectarine, fluorescent yellow, blue, or gray |
| Support material | Soluble support technology or breakaway support technology |
| Build volume | 254 × 254 × 305 mm (10 × 10 × 12 in.) |
| Layer thickness | Layer resolution low: 330 microns (0.013 in.) |
| | Layer resolution high: 254 microns (0.010 in.) |
| Dimensions and weight | 33 × 29 × 45 in and 326 lbs. |
| Regulatory compliance | CE/ETL |
| Approximately cost | 32.900 USD |

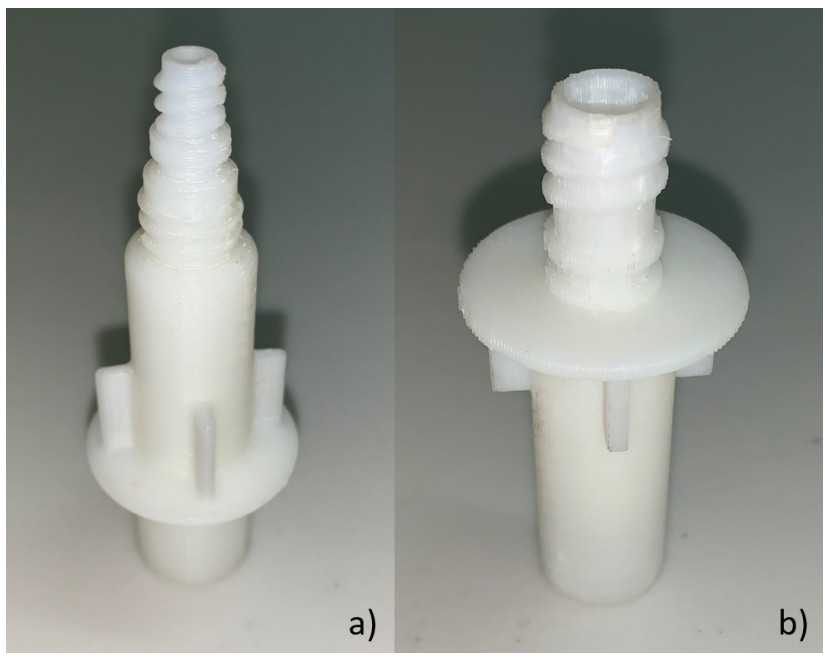

**Figure 5.** Acrylonitrile butadiene styrene (ABS) connector model: (**a**) BP plastic connector and (**b**) PB plastic connector.

Subsequently, the performance of the designs was evaluated by numerical analyses of the behavior of the fluid through BP and PB connectors under two study scenarios:

1. Fluid dynamic computational analysis in BP connector;
2. Fluid dynamic computational analysis in PB connector.

The connectors were manufactured in titanium, and the comparison was verified for the commercial brands. An experimental testbed was developed for the connectors to test whether the linking connections between the devices was airtightness. Equipment was used for detecting high leakage luminescence; there was no output of the fluid at the connection between the BP and PB connectors and the commercial devices at the inlet and the outlet flow connections where the fluid flows. The following section is divided by subheadings. It should provide a concise and precise description of the experimental results, their interpretation, as well as the experimental conclusions that can be drawn.

### 3.1. Fluid Computational Dynamics

The connectors were subjected to numerical simulation in the Solidworks® student version to simulate the fluid's path through the internal conduit. The boundary conditions are described in Table 2. The infusion liquid speed to the human body, and drainage of the dialysate solution from the human body are reported [17]. Furthermore, intraabdominal pressure was present during the treatment process [18].

**Table 2.** Boundary conditions for numerical simulation.

| Boundary Conditions | |
|---|---|
| **Connector Material** | **Plastic (PET)** |
| Fluid | Saltwater |
| Gravity | 9.81 $\frac{m}{s^2}$ |
| Infusion flow | 3.3333 $\frac{mL}{s}$ |
| Drain flow | 1.6666 $\frac{mL}{s}$ |
| Intraabdominal pressure | 5 mmHg |
| Temperature flow | 37.5 °C |

The flow starts with laminar behavior, but this becomes turbulent and significantly affects the fluid due to the pipe's boundary layer. Figure 6 shows the BP connector's connection points with the commercial elements and shows the thread features.

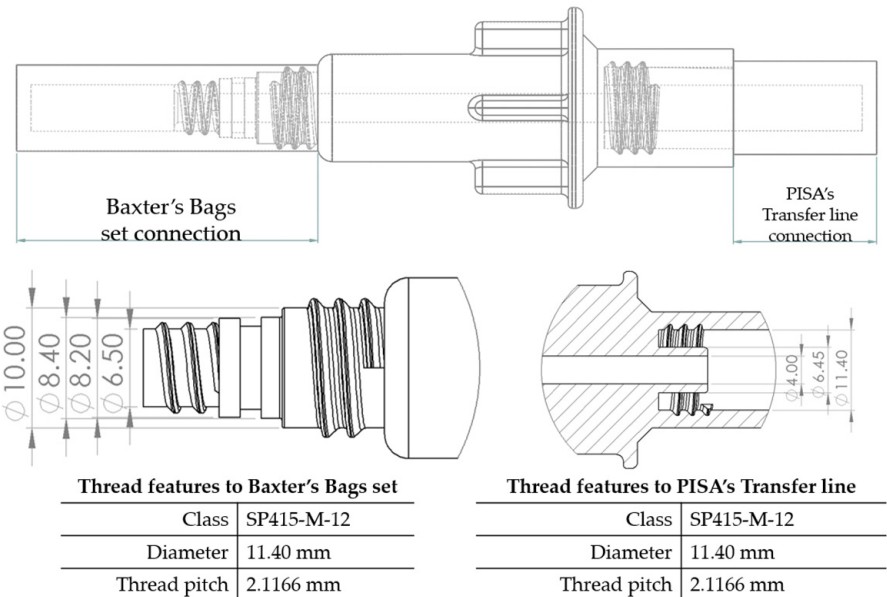

Baxter's Bags set connection

PISA's Transfer line connection

| Thread features to Baxter's Bags set | |
|---|---|
| Class | SP415-M-12 |
| Diameter | 11.40 mm |
| Thread pitch | 2.1166 mm |

| Thread features to PISA's Transfer line | |
|---|---|
| Class | SP415-M-12 |
| Diameter | 11.40 mm |
| Thread pitch | 2.1166 mm |

**Figure 6.** BP connector, join points, and thread features.

On the other hand, from the PISA bags connected to the Baxter transfer line, the PB connector was suitable for accomplishing the connection. Figure 7 shows a sketch of the connection points and thread features.

### 3.2. BP Connector in the Infusion Process

The length of the perfusion procedure from the bag with a dialysate solution to the human body is approximately 10 min. Table 3 shows the infusion parameters and numerical analysis of the fluid process in the BP connector.

It is observed that are no drawbacks when the fluid passes through the internal duct of this connector. Additionally, no vortices are generated in the flow. The speed profiles and pressure through the connector are acceptable to carry out a fluid exchange as is performed in the treatment of CAPD.

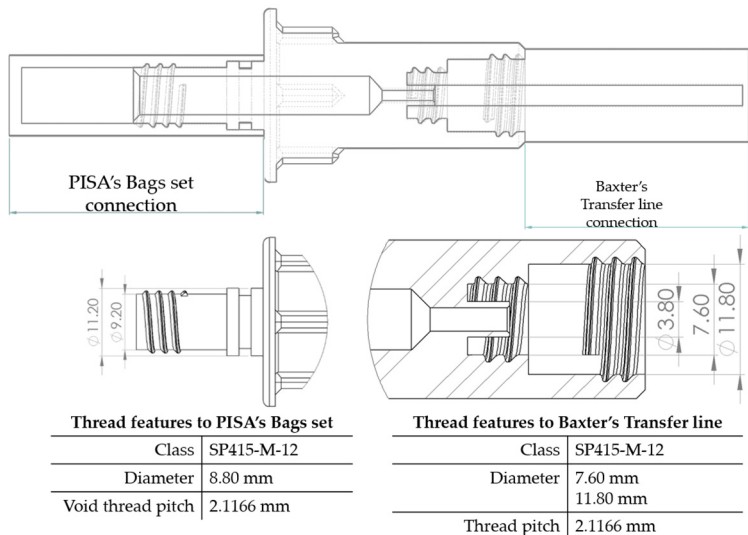

**Figure 7.** PB connector, join points, and thread features.

**Table 3.** Flow, pressure, and vorticity numerical analysis for BP connector—infusion fluid process.

| BP Connector—Infusion Fluid Process | |
| :---: | :---: |
| **Parameters** | **Numerical Analysis** |
| Flow (mm/s): simulation of BP connector flows evaluating velocity profiles starting around 3 mm/s and ending at $1.880380 \times 10^{-3}$ mm/s. | |
| Pressure (MPa): pressure was evaluated in the same connector when the fluid moved from the set of bags into the transfer line; results are closest between 0.000741 MPa and 0.000757 MPa. | |
| Vorticity ($s^{-1}$): the vorticity generated when the fluid is moved from the inlet flow connection and drained to the transfer line that reaches the human body was also analyzed. It is observed that the fluid rotation changes at the connection point, but the behavior shows values that do not generate complications, so the fluid leak at this point is null in the numerical simulation. | |
| Numerical simulation results of BP connector show adequate fluid behavior that does not present a risk to carrying out the fluid exchange. The pressure and vorticity remain stable, and it is ensured that the fluid profiles during infusion behave in a controlled manner without affecting the procedure of treatment. We can also observe the flow profiles at the linking point with the transfer trademark line. | |

### 3.3. BP Connector in the Draining Process

After the dialysis liquid is perfused into the peritoneal cavity, it remains within the human body for approximately 20 min, performing a cleaning process in the peritoneal cavity. When the cleaning process is completed, the draining procedure begins and lasts approximately 20 min; thus, the flow of the discarded liquid is less compared to that which enters. Table 4 shows the fluid draining process results.

**Table 4.** Flow, pressure, and vorticity numerical analysis for BP connector— fluid draining process.

| BP Connector—Drain Fluid Process | |
|---|---|
| **Parameters** | **Numerical Analysis** |
| Flow (mm/s): flow draining behavior into the bag to store the discarded liquid. The flow begins with a constant speed, and at the moment it crosses the connection point, the speed profiles increase towards the link with the bag set. |  |
| Pressure (MPa): the results of numerical simulation of the conduit pressure are negative and tend to increase when the flow goes through to the BP connector. |  |
| Vorticity ($s^{-1}$): the vorticity values increase when the liquid in the drainage stage reaches the linking point with the BP coupling connector. This behavior is due to the difference in geometries between the developed connectors and the transfer line of the trademark. At the linking point, the BP connector has a round area towards the internal conduit; this was designed to help the fluid enter. The human body does not generate the uncontrolled fluid rotation of the draining liquid. |  |
| Changes in the flow profiles are admissible because there is a controlled increase in the linking bags; the maximum value increases due to the change in the geometry of the ducts. The flow values are stable without leaks in the connections; this avoids pathology risk, such as peritonitis. |  |

The connector's linking point produced a flow change and increased it, letting the draining occur in a short time. At the point of linking between the elements, the speed of the fluid increases, favoring the treatment since it aims to reduce the time in which the liquid is discarded. This will speed up the completion of the treatment and provide the patient freedom to carry out their daily activities.

### 3.4. PB Connector in the Infusion Process

A PB connector numerical simulation was performed under the same border conditions; it is essential to point out that there is a reduction in the internal conduit. The reduction is from an internal diameter of 6.50 to 2.50 mm; this reduction is required to link

the outlet flow connection to the Baxter® transfer line. The infusion process for the PB connector is shown in Table 5.

**Table 5.** Flow, pressure, and vorticity numerical analysis for PB connector—infusion fluid process.

| PB Connector—Infusion Fluid Process | |
| --- | --- |
| **Parameters** | **Numerical Analysis** |
| Flow (mm/s): fluid behavior increases from $1.0 \times 10^{-7}$ to $4.496 \times 10^{-1}$ mm/s in the last part of the linking point. The duct neck increases through the reduction to 6 mm/s when the liquid is in the perfusion process. | 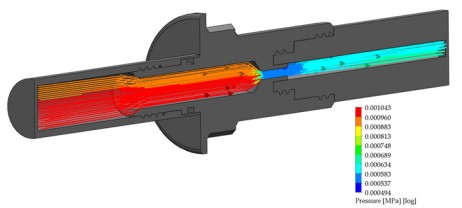 |
| Pressure (MPa): pressure results in this connector decreasing from 0.00416 MP at the bag linking input to 0.000634 MPa at the commercial transfer line output. The duct pressure presenting the reduction is constant at 0.000583 MPa and does not represent fluid transfer complications. | 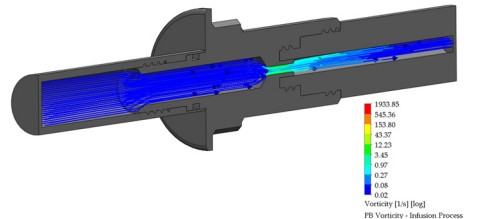 |
| Vorticity ($s^{-1}$): the vorticity connector behavior changes when it reaches the conical reduction; it presents a considerable increase up to 153.80 $s^{-1}$, and when it through into the conduit, it decreases balancing the fluid rotation until it reaches the link with the transfer line. | 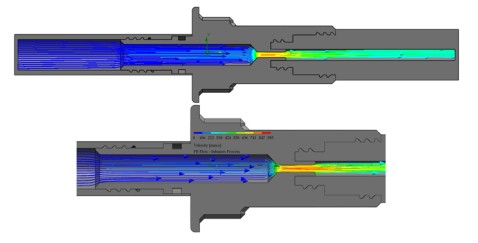 |
| Flow profiles obtained during infusion in the PB connector show that the profiles change when it reaches the conical reduction; it is seen that speed increases. When the fluid passes through the reduced duct, it regulates the velocity with a constant value towards the commercial transfer line; the perfusing flow will maintain in a permanent state until the liquid drains. | |

### 3.5. PB Connector in the Draining Process

As already mentioned, the draining process takes approximately 20 min, so the flow tends to be slower than the infusion process. The reduction inside this connector allows increasing the flow through the reduced duct, favoring the draining process. Table 6 shows the results for this stage.

The flow behavior, pressure, and vorticity are explicit in both connectors; these results favor continuing with the methodology proposed. Table 7 presents the numerical simulation behavior value during the infusion process into the connectors, comparing the results from the inlet flow connection (system bags) to the outlet flow connection (transfer line).

**Table 6.** Flow, pressure, and vorticity numerical analysis for PB connector—fluid draining process.

| PB Connector—Drain Fluid Process | |
|---|---|
| **Parameters** | **Numerical Analysis** |
| Flow (mm/s): flow behavior during the draining process receives a gradual increase in the profiles, and although only a reduction occurs, this allows an increase in the speed of emptying the peritoneal fluid from the transfer line. Numerical simulation shows the fluid's behavior when the waste liquid is drained from the corresponding bag's transfer line. | 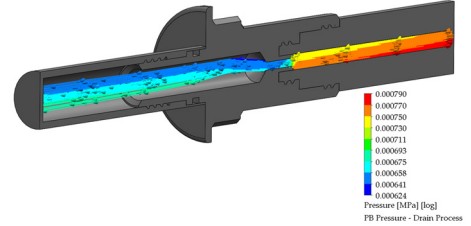 |
| Pressure (MPa): pressure presents a change, starting with a high-pressure that decreases when it crosses the PB connector reduction, reaching the bag assembly's connection and specifically the drain connection. | 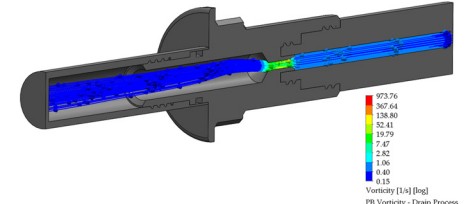 |
| Vorticity ($s^{-1}$): fluid rotation in the connector presents a change at the reduction, increasing values till 138.80 $s^{-1}$. However, once it crosses the reduced duct, it returns to the stable and continuous behavior that it initially presents. | 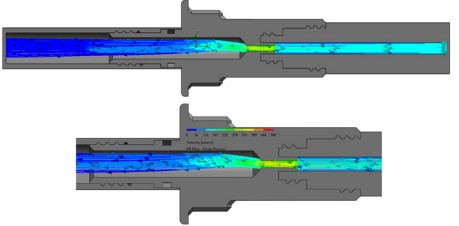 |
| Flow profiles in the draining process are acceptable, speed increases in the conical reduction and it is constant behaviour through the connector. The profiles increase with maximum speed movement at the reduction. | |

**Table 7.** Flow, pressure, and vorticity behavior of the fluid in the BP and PB connectors in the infusion process.

| Infusion Fluid Process | | | | | | |
|---|---|---|---|---|---|---|
| | **Flow (mm/s)** | | **Pressure (MPa)** | | **Vorticity ($s^{-1}$)** | |
| **Parameters** | **INLET BAG** | **Outlet Transfer Line** | **Inlet Bag** | **Outlet Transfer Line** | **Inlet Bag** | **Outlet Transfer Line** |
| BP connector | 3 | $1.605 \times 10^{-3}$ | 0.0009 | 0.0005 | 41.19 | 3.22 |
| PB connector | $1.650 \times 10^{-6}$ | 6 | 0.000416 | $1.608 \times 10^{-12}$ | 0.08 | 0.97 |

Table 8 presents the results obtained from the fluid drainage process numerical simulation through the BP and PB connectors; the results are classified from the peritoneal cavity's inlet flow to the bag set's outlet flow connection.

**Table 8.** Flow, pressure, and vorticity behavior of the fluid in the BP and PB connectors in the drainage process.

| | Drain Fluid Process | | | | | |
|---|---|---|---|---|---|---|
| | Flow (mm/s) | | Pressure (MPa) | | Vorticity (s$^{-1}$) | |
| Parameters | Inlet Peritoneal Cavity | Outlet Drainage Bag | Inlet Peritoneal Cavity | Outlet Drainage Bag | Inlet Peritoneal Cavity | Outlet Drainage Bag |
| BP connector | $1.301 \times 10^{-4}$ | $1.420 \times 10^{-3}$ | 0.00078 | 0.00075 | 1.06 | 0.42 |
| PB connector | $1.709 \times 10^{-4}$ | $1.429 \times 10^{-5}$ | 0.00075 | 0.00067 | 1.06 | 0.40 |

Three-dimensional printing and fused-deposition modeling obtained two detailed prototypes to verify the assembly between the devices developed with trademarks. The 3D models allowed us to establish manipulation and to estimate the exchange of fluid leaks through the connectors. It is essential to point out that the prototypes obtained with this material and by this method made it possible to specifically evaluate and analyze the necessary criteria for the existing problem and follow-up on the corresponding stage.

### 3.6. Titanium Manufactured Connectors

The connectors were manufactured with titanium "Ti-6Al-4V-ELI", which is used to develop medical devices in the health sector. The connectors' main function in this material lies in accurately connecting the Baxter® and PISA® commercial devices. The threading of the connectors was manufactured with support for both internal and external threads, adapting them to establish a mixture of equipment from both brands. The connectors were manufactured based on the proposed detailed designs. Engineering drawings and sketches were developed for the manufacturing of BP and PB connectors to generate them in titanium alloy, followed by a four-step process in approximately six hours of manufacturing.

i.      5/8 "TIAL6V4" Titanium aluminum alloy bar is trimmed to a certain length for each connector in two pieces per connector (total four pieces in 1.5 h);

ii.     Cannula drill for internal threads on the pieces for the linking points with the commercial devices (1.5 h process);

iii.    Cannula drill for external threads on the pieces to join it in a single one with a forged thread technique (1.5 h process);

iv.     Forged impact to obtain full BP and PB connectors (1.5 h process). Figure 8 shows the BP connector manufactured with Ti-6Al-4V: Baxter® infusion and drain bags to PISA® transfer line connection. It is used for the linking between elements.

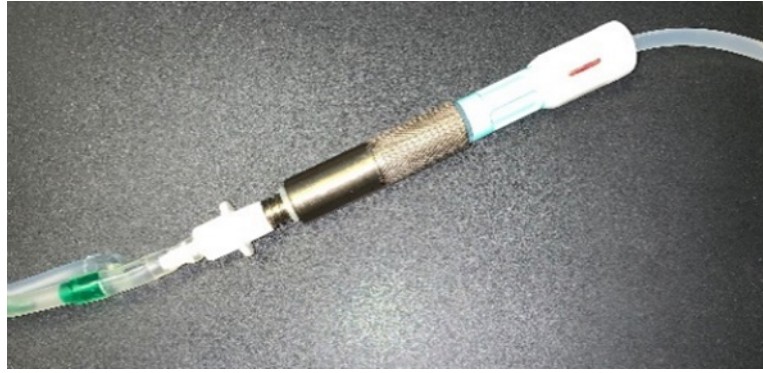

**Figure 8.** BP connector manufactured in titanium with branded equipment.

Similarly, Figure 9 presents the PB connector's manufacture that links the infusion bags and PISA® drainage to the Baxter® transfer line, obtaining a mix between equipment from both brands.

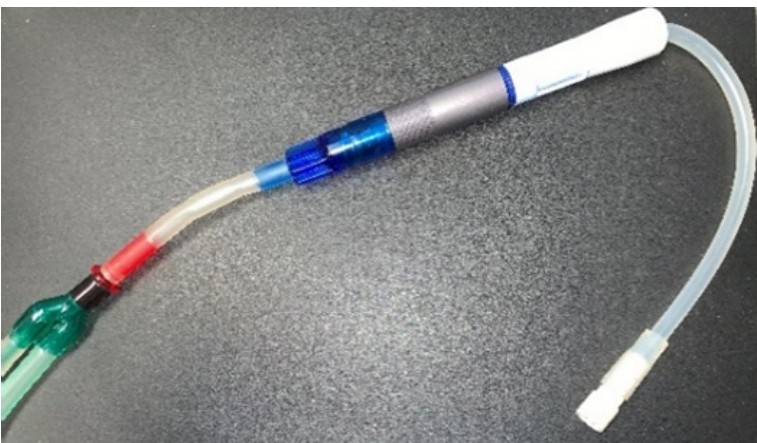

**Figure 9.** PB connector link manufactured in titanium with equipment from both brands.

The manufactured connectors do not present complications to the thread assembly with the two brands' equipment; it was also found that their handling is appropriate and can be used for the DPCA procedure following the corresponding process and operating instructions.

### 3.7. Experimental Testbed

The experimental connector validation was carried out by inspecting the airtightness in the areas at the linking join with the branded equipment through tests in which peritoneal fluid is recirculated through the BP and PB connectors are joined with the branded equipment. The experimental testbed simulates the CAPD procedure, which consists of dialysate flow from the set of bags to the transfer line. The following materials were considered for the manufacture and evaluation of the connectors:

- A 40 mm MoAS-type IPS aluminum profile bars;
- A peristaltic centrifugal pump;
- A high-luminescence leak detection kit.

The aluminum MoAs IPS bars allowed malleability in joining the BP and PB connectors in an ideal way to the branded products to perform the treatment as well as possible. The experimental testbed was adequate to visualize the fluid exchange configuration between the manufactured connectors and equipment from the most used brands in the Mexican health sector. The infusion bags were placed in a higher position for each connector used in the treatment of CAPD. A peristaltic pump was used to move the liquid through the connectors from the infusion bags to the drain bags. Figure 10 shows a diagram of the functionality and connection points among devices (commercial brands and BP and PB connectors).

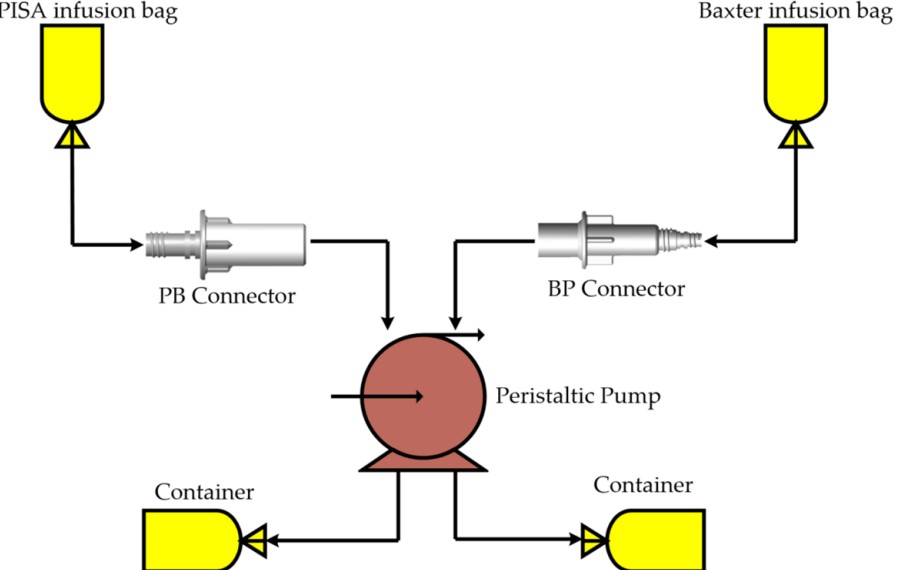

**Figure 10.** Titanium connectors linked with equipment from different brands.

In the procedure for detecting leaks at the connection points, the fluorescent compound was mixed with the dialyzing fluid bags to find leaks at the linking connections between devices. Figure 11 shows the testbed constructed with all the elements integrated.

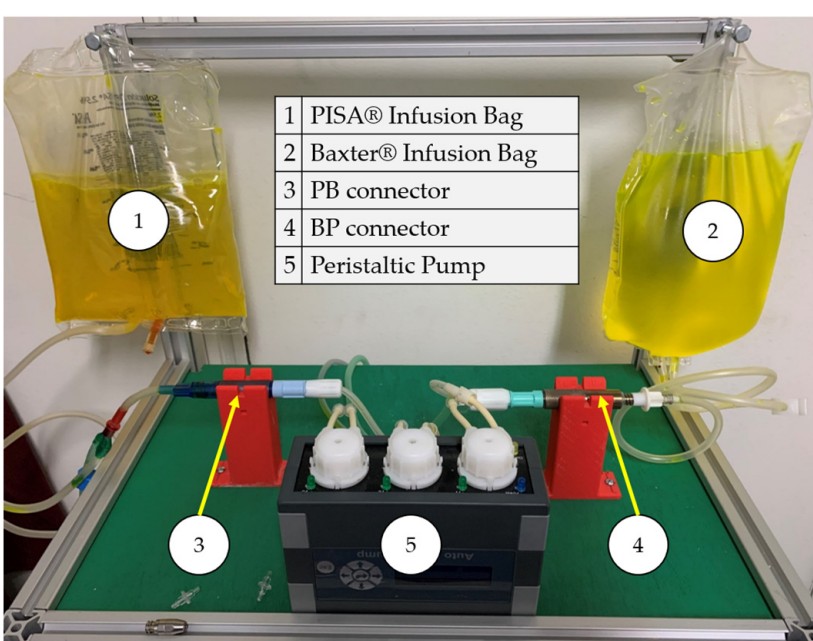

| 1 | PISA® Infusion Bag |
| 2 | Baxter® Infusion Bag |
| 3 | PB connector |
| 4 | BP connector |
| 5 | Peristaltic Pump |

**Figure 11.** Leak experimental bench for BP and PB connectors.

The connectors were placed on 3D-printed bases in the experimental testbed to better visualize the linked elements and the flow. Table 9 shows the configuration in the experimental bench and the evaluation of airtightness in each connector.

**Table 9.** Configuration of the experimental bench between the developed connectors and the commercial devices.

| Leak Experimental Essays for PB and BP Connectors | |
| --- | --- |
| **Description** | **Experimental Analysis** |
| PB connector placed on the 3D base connecting the PISA® infusion bag to the Baxter® transfer line. | 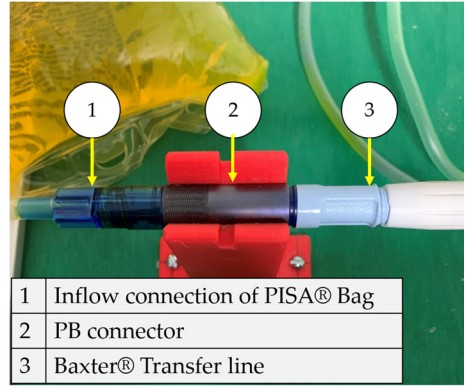 <br> 1  Inflow connection of PISA® Bag <br> 2  PB connector <br> 3  Baxter® Transfer line |
| The airtightness of the threaded evaluation connections was verified during the peristaltic pump's startup; the process evaluation involved constantly flowing the liquid for 5 min. Leak detection was performed in a dark room with a UV light lamp (inspection zones shown in red). | 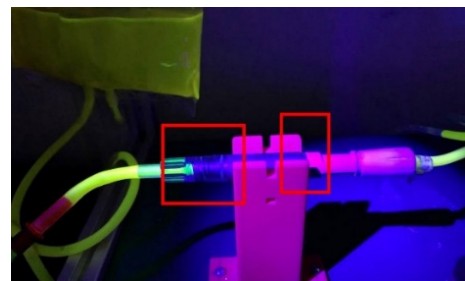 |
| BP connector placed on the 3D base connecting the Baxter® infusion bag to the PISA® transfer line. | 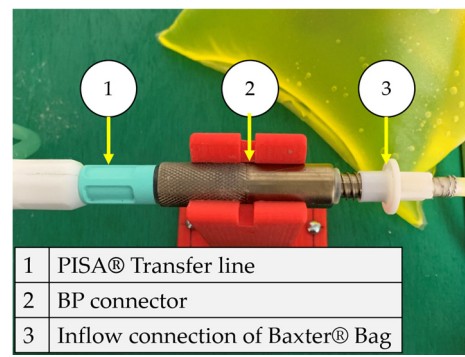 <br> 1  PISA® Transfer line <br> 2  BP connector <br> 3  Inflow connection of Baxter® Bag |
| Same airtightness tests were carried out for the BP connector, picturing the passage of the fluid over the connector illuminated by UV light (inspection zones shown in red). | 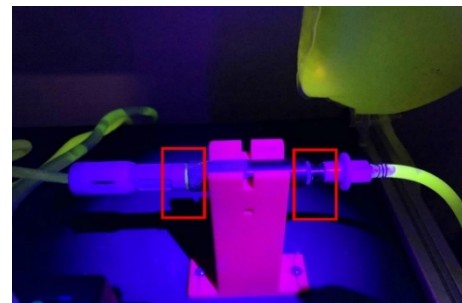 |

## 4. Discussion

The connectors present a specific geometry and ornament that is ideal for dealing with CKD in treatment. The connectors need to be operated from an ergonomic point of view for nurses and patient helpers. During the connection process of the BP and PB connectors with the commercial systems, it is essential to follow the hygiene rules established by each medical unit to carry out the treatment of CAPD, from the cleaning process to the handling

of each element for the exchange of fluids with the human body. Medical personnel have developed strategies to better use resources when they are insufficient, using sterilization methods in elements that allow their use for patients who require treatment in an emergency. Switching from one transfer line brand to another involves surgical intervention. BP and PB connectors are designed to provide an alternative to this problem, allowing immediate medical care for patients following the treatment protocol established.

It is essential to mention that both the shapes and geometry of each connector model (BP and PB) are correct for handling during the treatment process; this is an important issue in order to avoid critical complications of CKD such as infections. It is necessary to provide additional instructions to medical personnel or users to establish either BP or PB connectors with commercial systems to perform CAPD treatment.

The limitations that this research present are described in the following details:

- The results presented through the SolidWorks® student version computer program in this research demonstrate that the behavior of the fluid inside the designed connectors is constant, without presenting any abnormalities, even with the reduction in diameter in one of them. This allows us to validate the results obtained with another computer package that uses the finite element method (MEF), establishing a relationship between the results.

- Based on clinical analyses and knowledge of how to perform CAPD treatment, the conditions were established in the numerical simulation that the BP and PB connectors were subjected to. These determine the reliability of the results, which is acceptable. On the other hand, the visual study of the experimental testbed showed no fluid leak through the linking points.

- It would be ideal to use cell cultures for a final evaluation of the BP and PB connectors to support a determination of null bacteriological presence inside them, followed by in vitro tests to examine the infusion and drainage process of the treatment. NOM-152-SSA1-1996 provides all instructions for examining medical devices in PD, highlighting the importance of the cell cultures in plastic or metal alloy materials. The Comisión Federal para la Protección contra Riesgos Sanitarios (COFEPRIS) guidelines authorize the use of medical devices when they are developed in facilities with controlled environments and with a required hygiene grade.

The flow behavior inside the developed connectors does not present complications causing leaks of the fluid at the connection points. The behavior in medical-grade plastic or metal alloys is similar, so the employment of any connectors in the considered materials will carry out the process of treatment of DPCA satisfactorily, emphasizing that the handling is acceptable for each one. It is worth mentioning that the BP and PB connectors' flow profiles at the assembly points with transfer lines trademarks have increased flow at higher speeds, making a difference from the commercial flow systems.

There is little information about the use of a combination of connected devices to exchange fluids in the literature. Due to the lack of specific information and numerical simulation and experimental results, this is an alternative in the CAPD employing the BP and PB connectors' innovation.

This research has succeeded in realizing the intellectual protection of the connectors presented in this scientific report. It is essential to mention that these patents are authorized for use under the following codes: patent MX/2020/008489 and patent MX/f/2020/000314. It should be noted that the procedure required during CAPD begins with the connection of the Tenckhoff® catheter line to the dialysis bag. This multiprocessing will increase by one-step, which refers to connectors designed to generate the interchangeability of brands.

The certification of these new devices can be carried out by using an appropriate manufacturing company that has the ISO 9000 standard for acceptable manufacturing practices. The NOM-241-SSA1-2018 Good Manufacturing Practices for Medical Devices classifies those connectors as class II, as they are known in medical practice, and they may have variations in the material with which they are made or in their concentration generally introduced into the body in less than thirty days. If these two requirements are met, a

certificate can be requested from the corresponding Mexican entity called COFEPRIS to market it in Mexico's public and private health sector.

## 5. Conclusions

Continuous ambulatory peritoneal dialysis treatment is of the utmost importance, as it completely replaces the kidney function in the human body. It was possible to design two connectors that exchange the dialysate liquid, satisfactorily carrying out the treatment procedure with the two commercial brands used in Mexico. Baxter$^®$ and PISA$^®$ devices are made of polyvinyl chloride (PVC) and polyetheretherketone (PEEK), while the developed connectors were manufactured using a titanium alloy (medical-grade) and ABS, achieving a satisfactory connection between the elements and establishing a successful connection without leaks.

Currently, in Mexico, there is not a tool to enable the use of a combination of the two brands to perform the treatment of CAPD. The BP and PB connectors work as an element to mix the infusion and drainage bags of one brand with the transfer line of the other brand in order to accomplish this treatment. The assembly system in the developed connectors employs a particular thread, avoiding leaks of the fluid once the connection between the commercial devices with the BP and PB connectors is made. Additionally, a mechanical seal was added to ensure the connection was airtight in order to avoid pathological complications. Numerical tests allowed visual studios to analyze the fluid's behavior as it flowed through each connector. The BP connector flow profile patterns for the infusion process reached 3 mm/s at the inlet bag connection, with a controlled behavior of $2.201 \times 10^{-2}$ mm/s at the outlet transfer line flow of $1.605 \times 10^{-3}$ mm/s, which means a decrease in the velocity; in the draining process, maximum velocity profiles is $1.420 \times 10^{-3}$ mm/s at outlet drainage bag connection. The PB connector flow profile patterns in the infusion process are 6 mm/s at the outlet transfer line, which is the highest velocity in both cases; meanwhile, at the inlet bag, the connection decreased to $1.650 \times 10^{-6}$ mm/s

It is important to mention at the internal duct; the neck flow profiles reach 74 mm/s; in the drain, the process flow reaches $1.709 \times 10^{-4}$ mm/s at the inlet of the transfer line; at the internal duct neck in this connector, the velocity reaches 42 mm/s, and continue to $1.429 \times 10^{-5}$ mm/s to the outlet drainage bag connection. Favorable results were obtained, which allowed developing technical plans for the manufacturing of the connectors.

The connectors were manufactured in titanium "Ti-6Al-4V-ELI"; the linking was established through the thread between devices, suiting it successfully. Experimental leak detection evaluations in each connector and results determine the fulfillment of the objectives and the necessity under which they were developed. Thus, melted deposition technology is useful for considerable cost reduction, allowing observational and connection studies, which helped to design a detailed connector. The 3D models obtained in ABS accomplishing the connection and exchange of the fluid between the brands satisfactorily. Additionally, each model (ornament and geometries) was accurately modeled, which is why it was considered a highly efficient method for developing the connectors.

The immediate implementation of the BP and PB connectors will benefit a significant amount of Mexico's health sector, which helps around 8000 patients in institutions, as the Instituto de Seguridad y Servicios Sociales de los Trabajadores del Estado (ISSSTE) will perform the treatment with the transfer line that they are using. The experimental bench developed can be used to carry out tests with the drainage bags containing the liquid discarded by the human body; these tests would contribute to the research carried out so far.

The exchange of fluids between the infusion and drainage bags of one brand with another brand's transfer line was successfully achieved, considering the medical sector's connection criteria. The flow connector patterns show a considerable change due to the diameter geometry and reductions in each connector. However, these changes do not represent any complication in carrying out the procedure for CAPD treatment. From this proposal, it is possible to use the BP and PB connectors in specific tests involving

intervention with animals to carry out the exchange of fluids, verifying the connectors' functionality and validating their use in the treatment of CAPD using the standards of the Mexican organization COFEPRIS. The common connection procedure for treatment consists of correctly joining linking points between the transfer line (from the human body) to bags of the same brand under strict hygiene conditions. The use of the BP and PB connectors adds two steps that allow the use of the transfer line from one brand with bags from the other brand. Thus, the procedure involves (1) the correct handling and careful connection with the transfer line once the connection is successful; (2) the correct handling and careful connection with the bags under the same hygiene conditions.

Finally, it was determined that the developed connectors work correctly and can be manufactured in any suitable material for use in the health sector, strictly complying with the instructions for carrying out the treatment.

## 6. Patents

Patent MX/2020/008489 and patent MX/f/2020/000314 results from work reported in this manuscript.

**Author Contributions:** Conceptualization, C.R.T.-S. and M.A.G.-C.; methodology, C.R.T.-S.; software, M.A.G.-C.; validation, L.A.A.-P. and J.C.P.-R.; formal analysis, J.C.P.-R.; investigation, C.D.l.C.-A.; resources, C.R.T.-S.; data curation, J.C.P.-R.; writing—original draft preparation, M.A.G.-C.; writing—review and editing, C.R.T.-S.; visualization, J.C.P.-R.; supervision, C.R.T.-S.; project administration, C.D.l.C.-A.; funding acquisition, C.R.T.-S. All authors have read and agreed to the published version of the manuscript.

**Funding:** This research received no external funding.

**Institutional Review Board Statement:** This study did not require ethical approval.

**Informed Consent Statement:** Not applicable.

**Data Availability Statement:** Not applicable.

**Acknowledgments:** The authors thank the Consejo Nacional de Ciencia y Tecnología (CONACyT) and the Instituto Politécnico Nacional for the support received in the 20201964, 20200930 and 20210282 projects, as well as the support project "Proyectos de Desarrollo Tecnológico o Innovación para alumnos del IPN 2019", and finally the EDI grant, all from SIP/IPN. We appreciate the support of the specialists in nephrology José Ocotitla-Hernández and Dra. María Guadalupe Suárez López for the orientation in clinical subjects of CAPD who's affiliations are Subdirección de Regulación y Atención Hospitalaria, and Servicio de Diálisis Peritoneal, Hospital General Dario Fernández Fierro at Instituto de Seguridad y Servicios Sociales de los Trabajadores del Estado (ISSSTE).

**Conflicts of Interest:** The authors declare no conflict of interest.

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
