# Peer review of "Biomechanics Perspective for the Design and Manufacture of Continuous Ambulatory Peritoneal Dialysis Connectors"

_applsci, doi:10.3390/app11041502_

Round 1

Reviewer 1 Report

Grave-Capistrán et al. developed a customed connector in polymers and metals, via 3D printing, that enables the use of transfer lines and dialysis bags with different configurations and brands. Such connector may relieve the burden of patients with CKD, as it allows the flexible exchange between the products of the two brands in continuous ambulatory peritoneal dialysis. Overall, this manuscript seems not suitable for publication, and it needs to address several major concerns.

  • There is no experimental/method section, thus leading to a lack of experimental and simulation details. For example, the type of 3D printer and printing parameters are not provided; the printing resolution and capability are not characterized; the simulation results are not verified by published or experimental ones.
  • The language is rife with ambiguous descriptions and grammar errors, thus requiring significant improvement. For example, on page 1 line 16, “for exchage”, line 19 “an infusion and drain bags”. Language service provided by the MDPI and others may be considered.
  • Finally, the authors should discuss any issue of undermining intellectual property and healthcare regulation, as they attempt to modify the use of commercialized and, most likely, certified medical instruments.

Author Response

REPLY TO REVIEWER #1

Ms. Ref. No.: Applied Sciences 1037515

Title: Biomechanics perspective for design and manufacture of Continuous Ambulatory Peritoneal Dialysis connectors

Journal/Conference: MDPI-Applied sciences

Date of reply: 15/01/2021

Author for correspondence: Christopher René Torres San Miguel

Email address: [email protected]

We thank the reviewers for their valuable remarks.

Please, find below answers to the reviewers’ comments with reference to reviewers’ comments sets.

REVIEWER #1

Grave-Capistrán et al. developed a customed connector in polymers and metals, via 3D printing, that enables the use of transfer lines and dialysis bags with different configurations and brands. Such connector may relieve the burden of patients with CKD, as it allows the flexible exchange between the two brands' products in continuous ambulatory peritoneal dialysis. Overall, this manuscript seems not suitable for publication, and it needs to address several major concerns.

  • There is no experimental/method section, thus leading to a lack of experimental and simulation details. For example, the type of 3D printer and printing parameters are not provided; the printing resolution and capability are not characterised; published or experimental ones do not verify the simulation results.

Authors’ reply:

We thank this observation; a diagram has been included in figure 10 that works as an interpretation of the fluid from the beginning point through the BP and PB connectors to establish a connection with the commercial elements. Table 2 shows the main boundary conditions to generate a flow simulation in the connectors. Also, there is information about join points and threads for numerical simulation in Figure 6 and Figure 7.

Line 197 mention the 3D printer followed by a table (Table 1) that shows the printer features for three-dimensional modelling and ABS supplies, it is important to mention that FDM technology was used with a high layer printing resolution of 254 micron (0.010 in.). Due to the lack of information and researches talking about a juxtaposition with medical devices we cannot compare results at all but some numerical simulations in CAPD process such numerical results by Bengt, Gunna and Borje are related, they performed computational simulations of the peritoneal fluid flow in the CAPD, the fluid volume values increase continuously and profiles flow behaviour are controlled.

  • The language is rife with ambiguous descriptions and grammar errors, thus requiring significant improvement. For example, on page 1 line 16, “for exchange”, line 19 “an infusion and drain bags”. Language service provided by the MDPI and others may be considered.

Authors’ reply:

We agree with observation, so a deep revision in a full document has been done according to the language for publication, important modifications about the structure of lines were corrected. Also, some words change in order to emphasise the manuscript technically.

  • Finally, the authors should discuss any issue of undermining intellectual property and healthcare regulation, as they attempt to modify the use of commercialised and, most likely, certified medical instruments.

Authors’ reply:

This research has succeeded in realising the intellectual protection of the connectors presented in this scientific report. It is essential to mention that these patents are authorised for use under the following codes Patent MX/2020/008489 and Patent MX/f/2020/000314. It should be noted that the procedure required during CAPD begins with the connection of the Tenckhoff® catheter line to the dialysis bag. This multiprocessing will increase by one step, which refers to connectors designed to generate interchangeability of brands.

The certification of these new devices can be carried out by designating the appropriate manufacturing company, which has the ISO 9000 standard for acceptable manufacturing practices. The NOM-241-SSA1-2018, Good Manufacturing Practices for Medical Devices, classifies those connectors as class II known in medical practice and may have variations in the material with which they are made or in their concentration generally introduced into the body in less than thirty days. If these two requirements are met, the certificate can be requested from the corresponding Mexican entity called COFEPRIS to market it in Mexico's public and private health sector.

Reviewer 2 Report

The authors are conducting computational flow dynamics during different stages of continuous ambulatory peritoneal dialysis.  They have also constructed a experimental test bed to verify findings. This is a very important contribution in the field of applied biomedical devices   however the article needs to be entirely restructured in order to presented as a research article. 

1) The introduction is incomplete. 

a) The authors need to elaborate on the different types of dialysis and widen the statistics to international sources.  Aside from GFR what are the other clinically monitored for CAPD?  Even if they are not used they should be mentioned.  What is the acceptable range for GFR. 

b) The design requirement section in the main text needs to be moved to the introduction.

c) The specific aim of the paper is not clear.  The way it is written in the last paragraph of the introduction the authors have 3D printed the connectors (which they have not).  What about the titanium alloy?

 Did they manufacture both types plastic and metal and if yes how?

2) Materials and Methods and Results

a) A figure needs to be added in order to illustrate the experimental roadmap.

As it is lot of work has been done but it is difficult to follow the flow.

b) Going back to the specific aims, if any material has been manufactured or coated the methodology needs to be described.  

c) Sections 2.2 and 2.3 can be moved to supplementary materials.  The relevance of the geometry is outlined in the CFD analysis.

d) Figure 5 is a very nice figure, it needs to be moved up to establish the relevance of the CFD simulation to the biomedical application.

e)The numerical simulation figures should be grouped into series of snapshots as typically presented in research papers.  If grouping is not conducted they should be moved to supplementary materials.

f) The same applies to the additional testbed figures.  They need to be grouped or moved to the supplementary information.

g) How did the CFD simulation address leakage?  How can you relate the testbed results to the CFD?

3) Discussion

How do your results compare to other researchers?  

How are these results clinically relevant?

You need to cite references to prove your contribution.  Are there any clinical trials in place using some of these designs that you are proposing with associated data?

4) Conclusion

The conclusion has no quantitative summary of the CFD.  

5) The abstract needs to be rewritten including quantitative evidence.

Author Response

REPLY TO REVIEWER 2

Ms. Ref. No.: Applied Sciences 1037515

Title: Biomechanics perspective for design and manufacture of Continuous Ambulatory Peritoneal Dialysis connectors

Journal/Conference: MDPI-Applied sciences

Date of reply: 15/01/2021

Author for correspondence: Christopher René Torres San Miguel

Email address: [email protected]

We thank the reviewers for their valuable remarks.

Please, find below answers to the reviewers’ comments with reference to reviewers’ comments sets.

REVIEWER #2

The authors are conducting computational flow dynamics during different stages of continuous ambulatory peritoneal dialysis.  They have also constructed a experimental test bed to verify findings. This is a very important contribution in the field of applied biomedical devices   however the article needs to be entirely restructured in order to presented as a research article. 

  • The introduction is incomplete:
  1. The authors need to elaborate on the different types of dialysis and widen the statistics to international sources. Aside from GFR what are the other clinically monitored for CAPD? Even if they are no used they should be mentioned. What is acceptable range for GFR.
  2. The design requirement section in the main text needs to be moved to the introduction
  3. The specific aim of the paper is not clear. The way it is written in the last paragraph of the introduction the authors have 3D printed the connectors (which they have not). What about the titanium alloy?
  4. Did they manufacture both types plastic and metal and if yes how?

Authors’ reply:

  1. As mentioned, in-depth research has been done, lines from number 43 to 57 explain the APD modality in a summary way, and lines 87 to 94 contain complementary information about CAPD which is important in this manuscript. The acceptable range for GFR in men is around 130 ml/min per 1.73m2 of body surface area, and 120 ml/min in women, although there is no to the patient's age, gender, and condition.
  2. This is an important observation talking about the structure of the paper we appreciate it. Lines from 79 to 107 include detailed information about the requirement section as part of the introduction.
  3. We change the structure in the last paragraph of the introduction talking about the essential information, including the research's specific aims, including the research's specific aims, including the research's specific aims, lines from 95 to 106 explain it. Thereby that 3D modelling technology was transcendental in order to get a detailed design and functional titanium prototypes.
  4. Yes, both connectors (BP and PB) were manufactured. In the first stage in ABS plastic material, Table 1 shows the 3D printer model and features in order to get 3D preliminary prototypes which was an important step to get a detailed design. On the other hand, engineering drawing and sketches were developed for BP and PB connectors manufacture to generate them in titanium alloy followed by a four steps process in a six hours’ manufacture approximately; lines from 360 to 369 explain how to get in a summary way the manufacture for each connector.
    • Materials and Methods and Results
  5. A figure needs to be added in order to illustrate the experimental roadmap.

As it is lot of work has been done but it is difficult to follow the flow.

  1. Going back to the specific aims, if any material has been manufactured or coated the methodology needs to be described
  2. Sections 2.2 and 2.3 can be moved to supplementary materials. The relevance of the geometry is outlined in the CFD analysis.
  3. Figure 5 is a very nice figure, it needs to be moved up to establish the relevance of the CFD simulation to the biomedical application.
  4. The numerical simulation figures should be grouped into series of snapshots as typically presented in research papers. If grouping is not conducted they should be moved to supplementary materials.
  5. The same applies to the additional testbed figures. They need to be grouped or moved to the supplementary information.
  6. g) How did the CFD simulation address leakage? How can you relate the testbed results to the CFD?

Authors’ reply:

  1. A figure has been included in order to explain the flow patterns in the experimental bench. Figure 10 include a diagram about the flow lines from commercial bags set through BP and PB connectors to each transfer line. It is important to mention that the peristaltic pump is able to recirculate from one bag to another one trough the connected devices.
  2. Figure. Titanium connectors linked with elements of different trademarks. (attached in the file)
  3. Lines from 360 to 369 explain how to get in a summary way the manufacture for both connectors.
  4. We agree that it is important to keep sections 2.2 and 2.3 in order to follow the background of the BP and PB connectors design.
  5. We thank this observation, and we follow the instruction, Figure 5 now Figure 2 has been moved above section 2.1, this way the aim of the paper is also in the figure (exchange of fluids between two different brands).
  6. Numerical simulation figures have been grouped into tables with a description of the simulation:

Table 3. Flow, pressure and vorticity numerical analysis in BP connector – Infusion fluid process.

Table 4. Flow, pressure and vorticity numerical analysis in BP connector – Drain fluid process.

Table 5. Flow, pressure and vorticity numerical analysis in PB connector – Infusion fluid process.

Table 6. Flow, pressure and vorticity numerical analysis in PB connector – Drain fluid process

  1. Also, testbed figures about leakage detection have been grouped in a table. Table 9 shows the figures and description.
  2. Numerical simulation has been modified at the linking points with added threads connections representing bags set and the commercial devices' transfer lines. Next figures show the configuration in BP and PB connectors.

Figure. BP connector. (attached in the file)

Figure. PB connector. (attached in the file)

The analyses show that are not leakage during the simulation (infusion and drain process), flow profiles do not overpass threads connections, so airtightness in both connectors is correct. It is also important to say that each connector has a slot for a mechanical seal on the experimental analyses present in a null way leakage. On this way, It can say that both methods (numerical and experimental) share and reach this work's aim. Due to the COVID 19 situation, it is impossible to get into the laboratory to complement it. Tough, a high-speed camera was contemplated to visualise the fluid flow in specific areas such as the Titanium connector's mechanical seal and threads.

The following figures show the device's connections at the linking points, and there are no leakage overpassing threads connections, from commercial bags set to commercial transfer lines.

Figure. Numerical simulation on PB connector. (attached in the file)

Figure. Numerical simulation on BP connector. (attached in the file)

In addition, the importance of Whitworth threads allows a successful connection between devices. Finally, there are minor changes in the numerical results, and we concluded it is because of the length of the additional connections.

  • Discussion

  1. How do your results compare to other researchers?
  2. How are these results clinically relevant?
  3. You need to cite references to prove your contribution. Are there any clinical trials in place using some of these designs that you are proposing with associated data?

Authors’ reply:

  1. a) The development of a unique connector by the Fresenius brand exists; however, it is not reported in the open literature. Similarly, an automatic portable dialysis machine, developed in Italy, exists, of which clinical use is not reported. Previous information shows that at least in the European and American continent, there are no scientific reports of the use of connectors employed in various brands' juxtaposition. For this reason, this research could not comparate with other scientific reports.
  2. b) It is important to mention that the connectors' design has been evaluated in experimental test mannequins by three aspects: grip, handling and fixation, during the process of teaching ambulatory peritoneal dialysis to medical and nursing staff.

This connector will be benefit patients by interchanging brands. As future work, it is proposed to carry out a descriptive, observational, comparative, prospective study that will be applied in an instrumented dummy, where the parameters of filling (volume of infusion), permanence (permanence time), drainage phase in the connectors developed to show studies previous to the clinical application will be evaluated.

    3. c) It is not easy to compare our research with other scientific articles related to CAPD and reported in the open literature. Nevertheless, two works could have a relation with what has been developed in this research. The first is "Computer simulations of peritoneal fluid transport in CAPD, BENGT RIPPE et al., Kidney International, Vol. 40 (1991), PP. 315—325", which consists of a numerically integrated the phenomenological equations that describe the net ultrafiltration (UF) flow existing across the peritoneal membrane. It can be found dialysis volume patterns in differents CAPD conditions.

The second is "Comparison of Symmetric Hemodialysis Catheters Using Computational Fluid Dynamics, Timothy W.I. Clark et al., J Vasc Interv Radiol 2015; 26:252–259", which consist of finding flow characteristics of three symmetric catheters were compared based on computational fluid dynamics (CFD) as they relate to catheter function.

In our research, the design, manufacture on ABS a titanium materials, CFD behaviour of the fluid through the connectors, and the connectors' experimental links to know the possible leaks. In other words, the authors support that in the open literature are not reference to compare.

A research project has now been approved and began on 1 January 2021. This will be carried out at the Hospital General Dario Fernandez, which consists of validating the connectors by taking samples for bacteriological culture. In order to continue with the production phase for the Mexican health sector Afterwards, it is estimated that this connector's development will be able to reach all of Latin America. The following figure shows the approval of the project. 

Figure. Project funding (attached in the file)

  • Conclusion

The conclusion has no quantitative summary of the CFD.

Authors’ reply:

Summary information about the numerical results is in lines from 540 to 546 in the manuscript. It contains flow profile patterns values during the simulation of the infusion and drain process.’

  • The abstract needs to be rewritten including quantitative evidence

Authors’ reply:

We follow this instruction and the abstract was rewritten from 160 words to 196 words, including information about the results on both numerical and experimental cases.

Reviewer 3 Report

This paper presents the design case and methodology of two connectors used in Continuous Ambulatory Peritoneal Dialysis (CAPD) treatment. 3D printing models of the designs were built, and numerical simulations were carried on to establish flow patterns through these designs. An experimental test-bed was designed to verify the connection between fabricated devices with the elements of the market brands. The results presented in the numerical and experimental comparisons demonstrated that these connectors could be used in the treatment of the CAPD. This is a well written and presented design work. All aspects of the numerical and experimental testing performed are presented in comprehensive and understandable manner. It seems that the output of this work could be proven very useful for applications in Mexico. Its acceptance for publication is proposed.

Author Response

REPLY TO REVIEWER #3

Ms. Ref. No.:  Applied Sciences 1037515

Title: Biomechanics perspective for design and manufacture of Continuous Ambulatory Peritoneal Dialysis connectors

Journal/Conference: MDPI-Applied sciences

Date of reply: 15/01/2021

Author for correspondence: Christopher René Torres San Miguel

Email address: [email protected]

We thank the reviewers for their valuable remarks.

Please, find below answers to the reviewers’ comments with reference to reviewers’ comments sets.

REVIEWER #3

  • This paper presents the design case and methodology of two connectors used in Continuous Ambulatory Peritoneal Dialysis (CAPD) treatment. 3D printing models of the designs were built, and numerical simulations were carried on to establish flow patterns through these designs. An experimental testbed was designed to verify the connection between fabricated devices with the market brands' elements. The results presented in the numerical and experimental comparisons demonstrated that these connectors could be used in the treatment of the CAPD. This is a well written and presented design work. All aspects of the numerical and experimental testing performed are presented in comprehensive and understandable manner. It seems that the output of this work could be proven very useful for applications in Mexico. Its acceptance for publication is proposed.

Authors’ reply:

We thank your comments for the manuscript, and we consider an extensive revision in the whole document to improve the language to transmit information correctly. Also, numerical simulation changed, keeping the same boundary conditions. As a result, minor changes on flow patterns.

Round 2

Reviewer 1 Report

I have read the authors' reply to my and others' comments. I do not think the reply and the revisions are sufficient to address the raised concerns. Thus, the updated manuscript cannot justify the acceptance. It is sad to find out that many errors and ambiguities, even though pointed out in previous comments, remain. For example, another reviewer and I asked the authors to add the Methods section, which is a well-accepted standard format and a core component of a research paper. The method section is critical to convince the reviewer and the readers that the reported work is technically sound and reproducible. In contrast, the authors failed to add this Method section.

Reviewer 2 Report

Dear Authors,

Your modifications significantly improved the quality of the submission.  It was a pleasure to review the revised copy.